# Analysis of Low-Temperature Magnetotransport Properties of *NbN* Thin Films Grown by Atomic Layer Deposition

**Sahitya V. Vegesna** [1,2,*]**, Sai V. Lanka** [1,2]**, Danilo Bürger** [3]**, Zichao Li** [4]**, Sven Linzen** [1]
**and Heidemarie Schmidt** [1,2,*]

[1] Leibniz Institute of Photonic Technology, 07745 Jena, Germany; sai.lanka@uni-jena.de (S.V.L.);
sven.linzen@leibniz-ipht.de (S.L.)

[2] Institute for Solid State Physics, Friedrich Schiller University Jena, 07743 Jena, Germany

[3] Department Back-End of Line, Fraunhofer Institute for Electronic Nano Systems, 09126 Chemnitz, Germany;
danilobuerger@gmx.de

[4] Institute of Ion-Beam Physics and Materials Research, Helmholtz-Zentrum Dresden-Rossendorf (HZDR),
01314 Dresden, Germany; zichao.li@hzdr.de

[*] Correspondence: sahityav.vegesna@leibniz-ipht.de (S.V.V.); heidemarie.schmidt@leibniz-ipht.de (H.S.)

**Abstract:** Superconducting niobium nitride (*NbN*) films with nominal thicknesses of 4 nm, 5 nm, 7 nm, and 9 nm were grown on sapphire substrates using atomic layer deposition (*ALD*). We observed probed Hall resistance (*HR*) ($R_{xy}$) in external out-of-plane magnetic fields up to 6 T and magnetoresistance (*MR*) ($R_{xx}$) in external in-plane and out-of-plane magnetic fields up to 6 T on *NbN* thin films in Van der Pauw geometry. We also observed that positive MR dominated. Our study focused on the analysis of interaction and localisation effects on electronic disorder in *NbN* in the normal state in temperatures that ranged from 50 K down to the superconducting transition temperature. By modelling the temperature and magnetic field dependence of the MR data, we extracted the temperature-dependent Coulomb interaction constants, spin–orbit scattering lengths, localisation lengths, and valley degeneracy factors. The MR model allowed us to distinguish between interaction effects (positive MR) and localisation effects (negative MR) for in-plane and out-of-plane magnetic fields. We showed that anisotropic dephasing scattering due to lattice non-idealities in *NbN* could be neglected in the *ALD*-grown *NbN* thin films.

**Keywords:** superconductor; atomic layer deposition; *NbN* thin films; magnetoresistance; Coulomb interaction constant; valley degeneracy





## 1. Introduction

Its well known that the Lorentz force in a magnetic field on a moving charge is a force at a right angle to the magnetic field vector and the velocity vector of the charge. Therefore, external and internal magnetic field force charges into a curved trajectory. This preferential deflection of the electrons from the scattering events at defects or scattering from adjacent wave functions leads to the appearance of Hall voltage or magnetoconductance. The action of magnetic forces on the transport properties of semiconductors, metals, and also of superconducting materials in the normal conducting regime [1–5] has been extensively studied.

In our earlier work [6,7], we measured and analysed the transport properties of semiconductor thin films (*3D*) and of semiconductor ultrathin films (*2D*) in an external in-plane and out-of-plane magnetic field. The resistance of the investigated semiconductors, e.g., magnetic conducting oxides, increased when the temperature decreased. We extracted the temperature-dependent physical constants, e.g., isotropic Coulomb interaction constant, isotropic valley degeneracy factor, and isotropic dephasing length, along with the anisotropic *sd* exchange interaction energy and anisotropic dephasing length in the semi-

conductor ultrathin films (*2D*). The Coulomb interaction constant in the magnetic oxides decreased when the temperature decreased.

In this work we measured and analysed the transport properties of type II superconductor ultrathin films (*2D*) in the normal conducting regime above $T_c$ in an external in-plane and out-of-plane magnetic field. The investigated type II superconductor was an *ALD*-grown ultrathin *NbN* film on sapphire substrate, which had been measured in Van der Pauw geometry. The resistance of the *NbN* thin films slightly increased when the temperature decreased down to $T_c$. We analysed the magnetotransport properties, i.e., the increase in resistance in a magnetic field (positive MR) and the decrease in resistance in a magnetic field (negative MR), of moving charges in the *NbN* ultrathin films (*2D*) by modelling the interaction effects that caused positive MR (Zeeman interaction, spin–orbit interaction) and localisation effects that caused negative MR (dephasing). So far, only the combined magnetoresistance of positive MR and negative MR has been analysed for *NbN* thin films (*3D*) in out-of-plane and not in in-plane applied magnetic fields [8,9]. The combined magnetoconductivity model does not distinguish between positive MR and negative MR by extracting physical parameters, such as spin–orbit scattering time and dephasing time or optional spin–orbit scattering length and dephasing length [8,9].

## 2. Materials and Methods

Plasma-enhanced atomic layer deposition (plasma-enhanced *ALD*) was used to prepare $1 \times 1 \, \text{cm}^2$ large superconducting niobium nitride (*NbN*) films with thicknesses of 4 nm, 5 nm, 7 nm, and 9 nm on sapphire substrates using an *ALD* growth rate of only 0.47 Å per cycle. The *NbN* thickness variation in the used $1 \times 1 \, \text{cm}^2$ large area was less than 1% because of the *ALD* plasma, which had been generated by a three-inch source in the *ALD* deposition chamber. The critical temperature ($T_c$) of ultrathin *NbN* films depends on the film thickness and is lower than the $T_c$ of bulk *NbN* ($T_c = 16.2 \, \text{K}$) [10]. The *NbN* films were polycrystalline and characterised by the cubic $\delta$-*NbN* phase, and small amounts of oxygen and carbon impurities were detectable. Detailed information about the *NbN* film's properties and the *ALD* process can be found in Refs. [11–13].

We analysed the temperature-dependent change of interaction and localisation in *NbN* in the normal state, which is slightly above the superconducting transition temperature ($T_c$). The superconducting transition temperature depends on stoichiometry and defects in the *NbN* thin films. In order to obtain different superconducting transition temperatures $T_c$, we varied the thickness of the NbN thin films analogous to Ref. [11] and analysed the temperature-dependent magnetotransport properties. *NbN* thin films with nominal thicknesses (*t*) of 4 nm, 5 nm, 7 nm, and 9 nm were prepared by atomic layer deposition on $1 \times 1 \, \text{cm}^2$ large sapphire substrates at a substrate temperature of 380 °C. The Hall resistance (*HR*) ($R_{xy}$) and magnetoresistance (MR) ($R_{xx}$) measurements were carried out using the Hall measurement system *LakeShore HMS 9700A* in an external magnetic field up to 6 T in temperatures that ranged from 50 K down to the superconducting transition temperature ($T_c$) with an applied current of 100 µA. Note that the enclosures in the used setup for Hall/MR measurements did not completely isolate the electronics from the magnetic fields.

### 2.1. Critical Temperature

We measured the temperature-dependent sheet resistivity ($\rho_{xx}$) and determined the critical temperature ($T_c$) as 8.77 K, 10.75 K, 11.82 K, and 12.72 K for *NbN* thicknesses of 4 nm, 5 nm, 7 nm, and 9 nm, respectively (see Figure 1). For the 4 nm-thick sample the transition was not completed down to 8 K; 8 K was the minimum achieved temperature inside the used Hall/MR setup. However, previous investigations carried out at lower temperatures and inside a magnetically shielded setup still showed a complete transition for small film thicknesses [11]. The incomplete transition was most probably due to magnetic field stray fields that arose from the measurement setup (<50 Gauss) [14]. In general, in this work we defined the critical temperature as the temperature where the sheet resistivity $\rho(T_c)$ dropped to 1 µΩcm. These results were in agreement with the work from Zhang et al. [15]

and observed decreases in $T_c$ with decreasing *NbN* thicknesses had also been observed for other superconducting materials. Ivry et al. [16] proposed different mechanisms underlying thickness dependence $T_c$ in superconducting thin films and analysed the thickness value where a superconducting to insulating transition (*SIT*) could be observed. *ALD*-grown *NbN* films showed the *SIT* for an *NbN* film thickness of 2.9 nm [11] and the thicknesses of the *NbN* thin films investigated in this work were close to the thickness where *SIT* was observed.

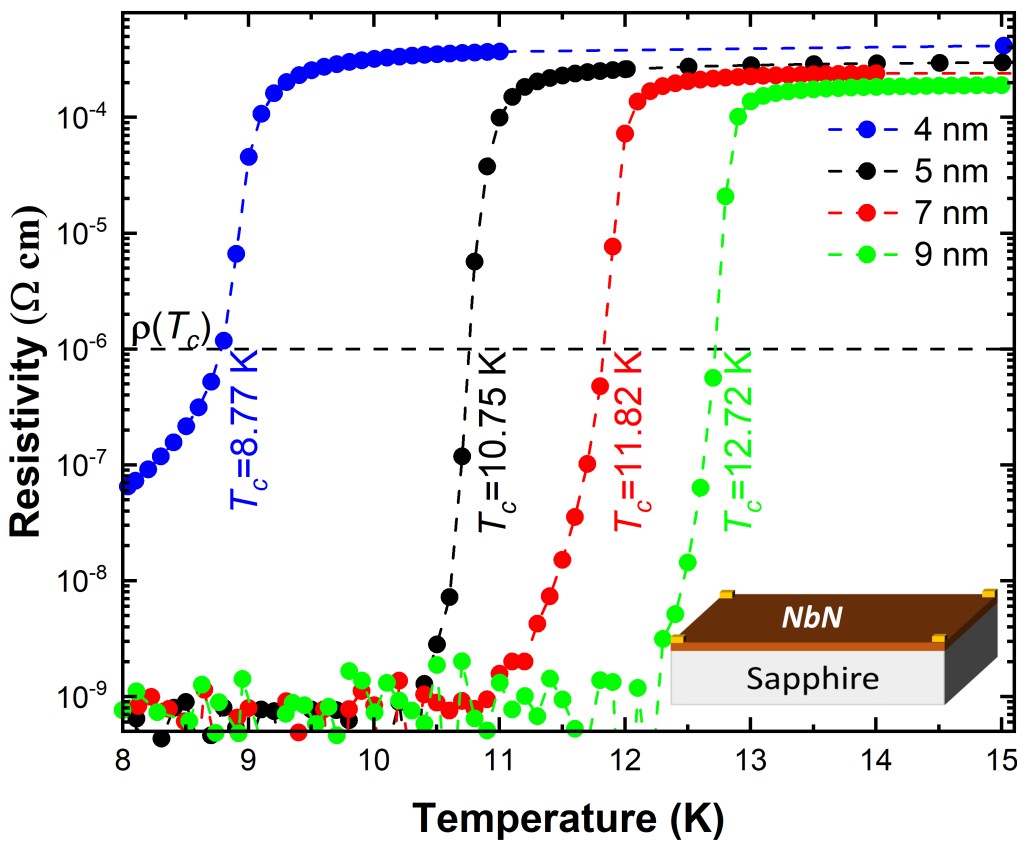

**Figure 1.** Temperature-dependent resistivity ($\rho$) of 4 nm, 5 nm, 7 nm, and 9 nm *NbN* on sapphire in Van der Pauw geometry. The critical temperature ($T_c$) was defined at $\rho$ = 1 μΩcm for generalisation. The critical temperatures were 8.77 K, 10.75 K, 11.82 K, and 12.72 K for 4 nm, 5 nm, 7 nm, and 9 nm *NbN* thin films, respectively.

*2.2. Hall Resistance*

Figure 2 shows the Hall resistance of 4 nm-, 5 nm-, 7 nm-, and 9 nm-thick *NbN* thin films in Van der Pauw geometry at temperatures above $T_c$ (Figure 2a–d, respectively). The *HR* offset at the zero field in Figure 2b,c was possibly due to a thermoelectric voltage related to Ettingshausen thermalisation effects during measurement [17]. The negative linear dependence of Hall resistance with magnetic field indicated that the *NbN* thin films were *n*-type conducting. As long as Hall resistance ($R_{xy}$) was linear, carrier concentration ($n$) and mobility ($\mu$) could be extracted from Equations (1)–(3) at normal conducting temperatures from the measured sheet resistance ($R_{xx}$) (Figure 3), respectively.

$$\frac{dR_{xy}}{dH} = \frac{1}{net} \tag{1}$$

$$\rho = R_{xx} \cdot t = \frac{1}{ne\mu} \tag{2}$$

$$\mu = \frac{dR_{xy}}{dH} \cdot \frac{1}{R_{xx}} \tag{3}$$

where $H$ is the magnetic field, $n$ is carrier density, $\rho$ is resistivity, $e$ is electron charge, $t$ is thickness, and sheet resistance is ($R_{xx}$). For temperatures 20 K, 30 K, and 50 K there were no significant changes in $n$ or in $\mu$. However, the carrier concentration decreased and the mobility in *NbN* thin films increased when the measurement temperature approached $T_c$. From mobility and high carrier concentrations in the orders of $10^{22}$ cm$^{-3}$, and from the temperature-dependent product $k_F l$ (Equation (4)) between the Fermi wave vector ($k_F$) and the mean free path ($l$) ($k_F l > 1$), one could conclude that the electrons in *NbN* thin films were weakly localised. The carrier concentration of bulk *NbN* amounted to $1.12 \times 10^{23}$ cm$^{-3}$ [15].

$$k_F l = \frac{\hbar(3\pi^2)^{2/3}}{e^2 \rho n^{1/3}} \tag{4}$$

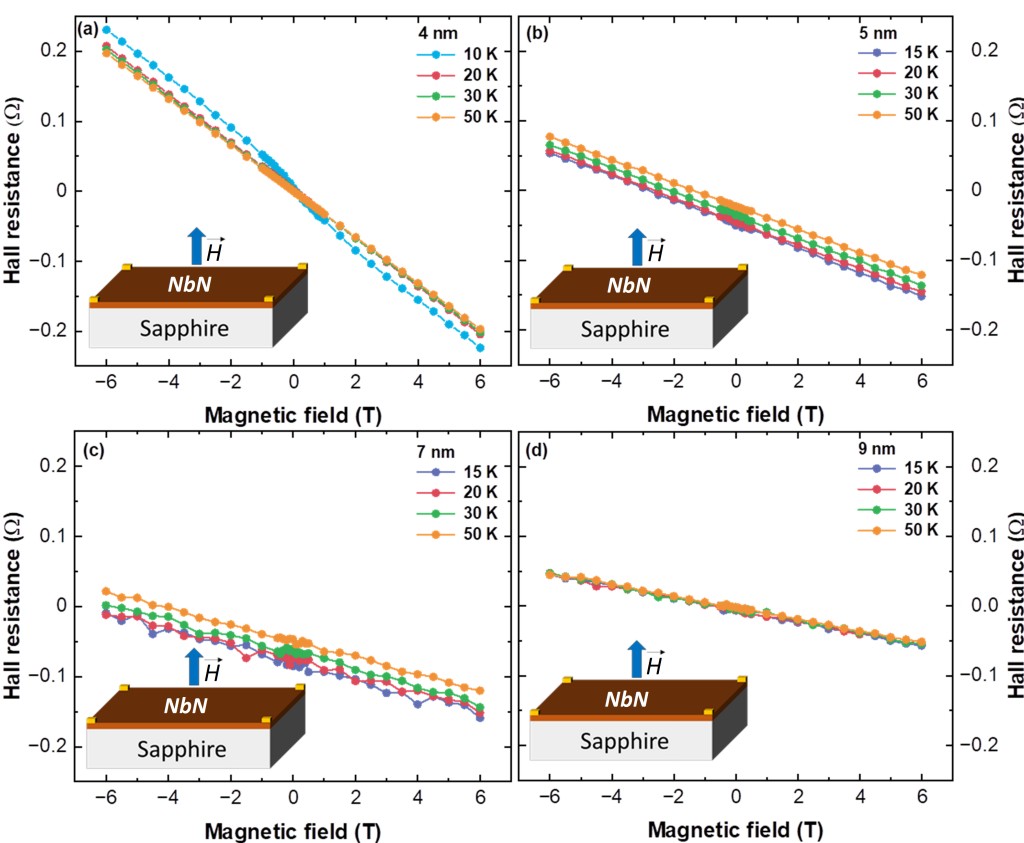

**Figure 2.** Hall resistance measured in Van der Pauw geometry at 10 K, 20 K, 30 K, and 50 K for (**a**) 4 nm, and at 15 K, 20 K, 30 K, and 50 K for (**b**) 5 nm, (**c**) 7 nm, and (**d**) 9 nm *NbN* thin films on sapphire. As expected, for a fixed temperature, the slope of magnetic-field-dependent Hall resistance of *n*-type conducting *NbN* thin films decreased with increasing thickness of the *NbN* thin films. Except for the Hall resistance of the 4 nm *NbN* thin film at 10 K, which was close to the critical temperature of the 4 nm-thick *NbN* thin film. All Hall resistance curves were also linear and in the magnetic field range −1 T to +1 T.

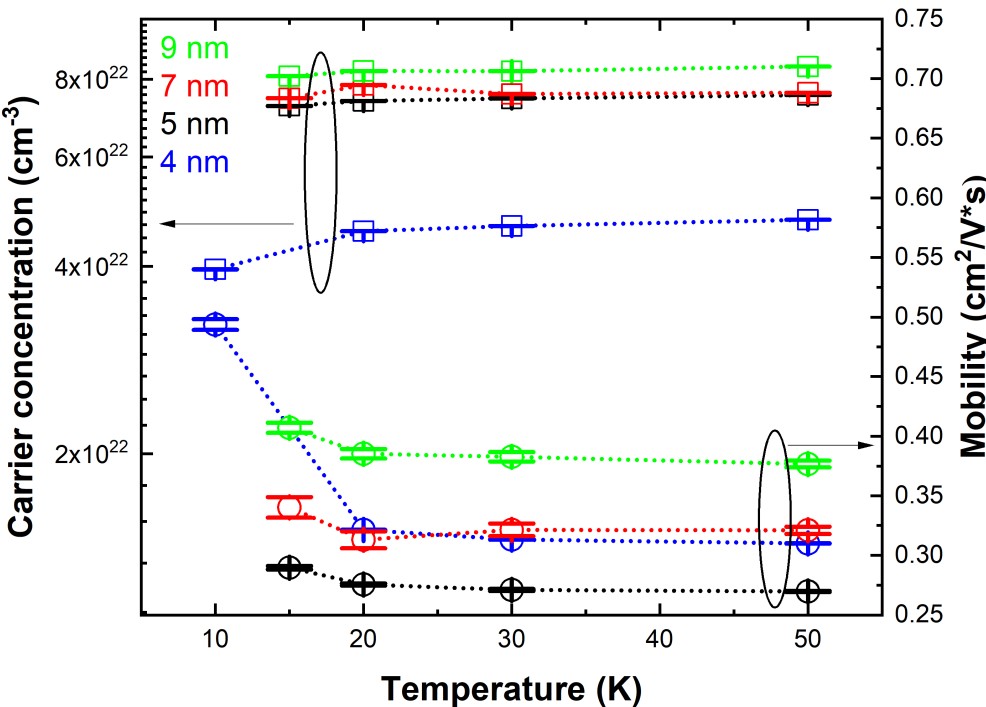

**Figure 3.** (**Left axis**) Temperature-dependent carrier concentration in cm$^{-3}$ and (**right axis**) mobility in cm$^2$/Vs of a 9, 7, 5, and 4 nm-thick *NbN* thin film on sapphire above the critical temperature. The carrier concentration and mobility of bulk *NbN* is also indicated. The measurement error is indicated by error bars and lies in the range $3.24 \times 10^{17}$ cm$^{-3}$ to $1.97 \times 10^{19}$ cm$^{-3}$ for carrier concentration and in the range $0.8 \times 10^{-4}$ cm$^2$/Vs to $86 \times 10^{-4}$ cm$^2$/Vs for mobility.

## 3. Results

The magnetoresitance MR at a temperature *T* is the relative change of resistivity in an applied magnetic field *H* (Equation (5)).

$$\text{MR} = \frac{\rho(H, T) - \rho(0, T)}{\rho(0, T)} = \rho(H, T) \times \Delta\sigma(H, T) \tag{5}$$

The sheet resistivity ($\rho = R_{xx} \times \textit{Thickness}$) is the intrinsic property of a material and is related to conductivity as $\sigma = \frac{1}{\rho}$. As will be shown in the following, the analysis of the changes in MR from $\Delta\sigma$ based on the the temperature-dependent transport mechanisms under varying magnetic fields gave a deeper insight into physical parameters such as the Coulomb interaction constant ($F_\sigma$), spin–orbit interaction energy ($\Delta E_{so}$), and dephasing length ($L_\phi$) of the NbN thin films. $F_\sigma$ is an isotropic temperature-dependent and magnetic-field-independent parameter that is related to the lifting of the degeneracy of electronic states in an external magnetic field (Zeeman splitting scattering). On the other hand, due to the small film thickness, magnetic-field-dependent localisation (dephasing scattering $L_\phi$) and interaction (spin–orbit scattering $\Delta E_{so}$) were anisotropic. We obtained MR measurements under magnetic fields applied in-plane (*i/p*) and out-of-plane (*o/p*) to the film to exploit the isotropic $F_\sigma$ and anisotropic $L_\phi$ and $\Delta E_{so}$ (Figure 4). The resistance of the *NbN* thin films on the insulating sapphire substrate was measured at normal conducting temperatures, between 10 K and 50 K in magnetic fields up to 6 T in Van der Pauw geometry. At normal conducting temperatures (above $T_c$) we observed positive magnetoresistance (Figure 4).

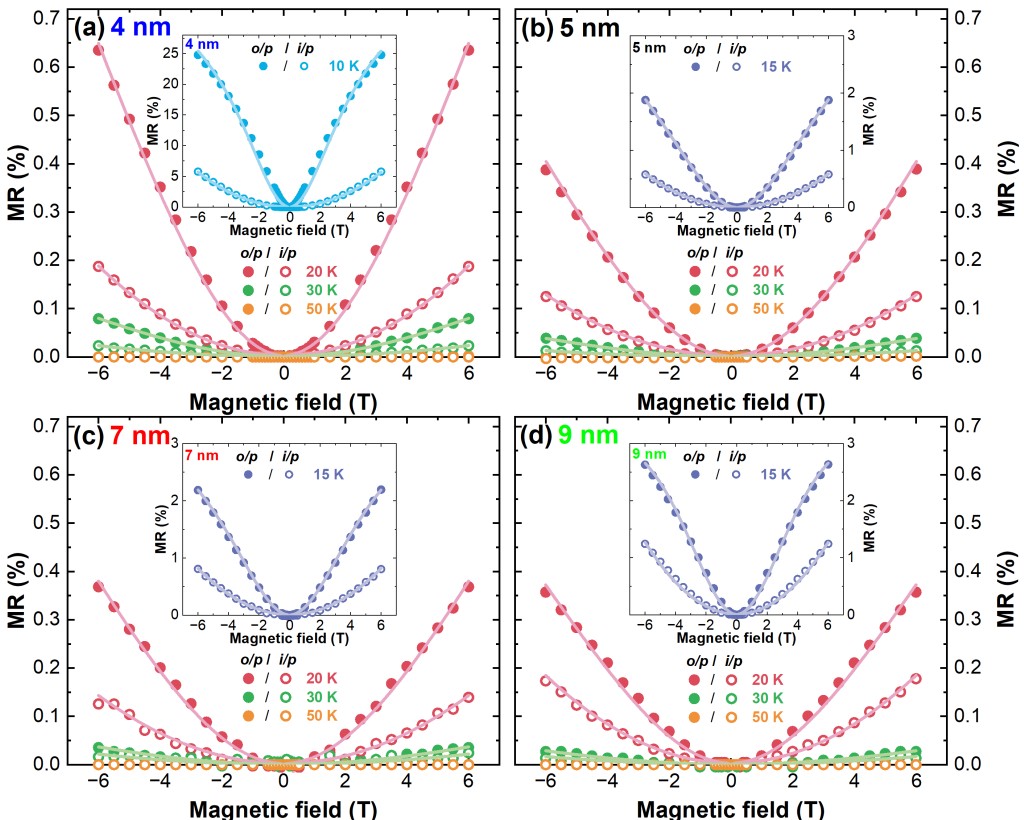

**Figure 4.** Measured and modelled out-of-plane (*o/p*, closed symbols) and in-plane (*i/p*, open symbols) magnetoresistance (MR) in % above $T_c$ at 10 K, 20 K, 30 K, and 50 K for (**a**) 4 nm, and at 15 K, 20 K, 30 K, and 50 K for (**b**) 5 nm, (**c**) 7 nm, and (**d**) 9 nm *NbN* thin films on sapphire in the magnetic field range from −6 T to +6 T. Sheet resistance measurements at 50 K revealed zero MR for all the samples. The magnetoresistance at 50 K lay below the measurement resolution limit ($3 \times 10^{-3}$%).

## 4. Discussion

There were two independent MR phenomena, one that explained exchange interaction effects ($\mathrm{MR}_I \propto \Delta\sigma_I$) and another that explained localisation effects ($\mathrm{MR}_L \propto \Delta\sigma_L$). $\mathrm{MR}_I$ and $\mathrm{MR}_L$ could be considered as independent effects and explicitly combined given $k_F l > 1$ [1] (Table 1).

**Table 1.** Temperature-dependent modelling parameters. The Coulomb interaction constant ($F_\sigma$ in kgm$^3$/s$^2$C$^2$) was extracted from in-plane (*i/p*) and MR was isotropic, i.e., $F_\sigma$(*i/p*) = $F_\sigma$(*o/p*). spin–orbit interaction energy ($\Delta E_{so}$ in meV), dephasing length ($L_\phi$ in nm), and valley degeneracy factor ($n_v$) were extracted from out-of-plane (*o/p*) MR for samples 4 nm , 5 nm, 7 nm, and 9 nm.

| Sample | Temperature K | $F_\sigma$ kgm$^3$/s$^2$C$^2$ | $\Delta E_{so}$ meV | $L_\phi$ nm | $n_v$ | $k_F l$ |
|---|---|---|---|---|---|---|
| | | *i/p* and *o/p* | *o/p* | *o/p* | *o/p* | *o/p* |
| 4 nm | 10 | 185 | 1.38 | 6.0 | 185 | 3.61 |
| | 20 | 19 | 1.07 | 4.6 | 1 | 2.58 |
| | 30 | 5 | 1.07 | 4.0 | 1 | 2.55 |
| 5 nm | 15 | 60 | 1.09 | 4.0 | 60 | 3.17 |
| | 20 | 22 | 0.95 | 4.0 | 1 | 3.06 |
| | 30 | 5 | 0.95 | 4.0 | 1 | 3.02 |
| 7 nm | 15 | 140 | 1.23 | 4.8 | 140 | 3.79 |
| | 20 | 42 | 0.79 | 4.8 | 1 | 3.61 |
| | 30 | 14 | 0.79 | 4.8 | 1 | 3.62 |
| 9 nm | 15 | 360 | 1.25 | 5.2 | 360 | 4.79 |
| | 20 | 90 | 0.85 | 4.9 | 23 | 4.60 |
| | 30 | 18 | 0.75 | 4.9 | 1 | 4.56 |

**Theorem 1.** *The exchange interaction and localisation effects can be studied from the change in magnetoconductivity, which is related to magnetoresistance as follows:*

$$\sigma(H,T) = \frac{1}{\rho(H,T)} \tag{6}$$

$$\Delta\sigma(H,T) = \Delta\sigma_I(H,T) + \Delta\sigma_L(H,T) \tag{7}$$

*The exchange interaction effects on magnetoresistance ($MR_I$) are positive and the field dependent change in conductivity for two and three dimensions (2D and 3D, respectively) can be modelled as follows [1]:*

$$\Delta\sigma_I(H,T) = \sigma_I(H,T) - \sigma_I(0,T) = -\frac{e^2}{\hbar}\frac{F_\sigma}{4\pi^2}g_2(\beta) \tag{8}$$

$$\Delta\sigma_I(H,T) = \sigma_I(H,T) - \sigma_I(0,T) = -\frac{e^2}{\hbar}\frac{F_\sigma}{4\pi^2}\sqrt{\frac{k_BT}{2\hbar D}}g_3(\beta) \tag{9}$$

*where e is the electron charge, $k_B$ is the Boltzmann constant, $F_\sigma$ is the screening parameter for the column interaction ranging between 0 and 1 (for 2D modelling $F_\sigma > 1$), and D is the diffusion constant. $g_2(\beta)$ [18] and $g_3(\beta)$ [19] are*

$$g_2(\beta) = \int_0^\infty d\Omega \frac{d^2}{d\Omega^2}[\Omega N(\Omega)]ln\left|1 - \frac{\beta^2}{\Omega^2}\right| \tag{10}$$

$$g_3(\beta) = \int_0^\infty d\Omega \frac{d^2}{d\Omega^2}[\Omega N(\Omega)] \times (\sqrt{\Omega+\beta} + \sqrt{\Omega-\beta} - 2\sqrt{\Omega}) \tag{11}$$

*where β is the ratio between the sum of Zeeman splitting energy ($g_e\mu_BH$), the spin–orbit exchange interaction energy ($\Delta E_{so}$), and the thermal energy, which is given as follows [6]:*

$$\beta = \frac{g_e\mu_BH + \Delta E_{so}L(x)}{k_BT}, \tag{12}$$

*where* x $= \frac{\mu_{eff}H}{k_BT}$ *and Langevin function* $L(x) = coth(x) - \frac{1}{x}$ . *The localisation magnetoresistance ($MR_L$) in Equation (13) and in Equation (14) is expressed as [1].*

$$\Delta\sigma_L(H,T) = \sigma_L(H,T) - \sigma_L(0,T) = \frac{n_v e^2}{2\pi^2\hbar}\left[\psi\left(\frac{1}{2} + y_\phi\right) - \ln y_\phi\right] \tag{13}$$

$$\Delta\sigma_L(H,T) = \sigma_L(H,T) - \sigma_L(0,T) = \frac{n_v e^2}{2\pi^2\hbar}\sqrt{\frac{eH}{\hbar}}f_3(y_\phi) \tag{14}$$

*for 2D and 3D, respectively. ψ is the digamma function, $n_v$ is the valley degeneracy factor, $y_\phi = \frac{\hbar}{4eHL_\phi^2}$, and*

$$f_3(y) = \sum_{n=0}\left[\left(2\sqrt{n+y+1} - \sqrt{n+y}\right) - \frac{1}{\sqrt{n+y+1/2}}\right]. \tag{15}$$

**Proof of Theorem 1.** The analysis of magnetoresistance in thin films is a well-established scientific practice to analyse the exchange interaction and localisation in weakly localised disordered electronic systems. These effects are independent phenomena and contribute to scattering in an external magnetic field as long as the product $k_Fl > 1$. $k_Fl$ of investigated *NbN* thin films were calculated from experimental sheet resistivity (ρ) and carrier concentration (n) extracted from Hall resistance measurements (Equation (4)) and were larger than 1 (Table 1).

Due to the small *NbN* film thickness, magnetic-field-dependent localisation from dephasing scattering and interaction from spin–orbit scattering was anisotropic. In ad-

dition, since the measured magnetotransport properties were anisotropic (Figure 4), i.e., MR($i/p$) ≠ MR($i/p$) the transport properties were two dimensional (*2D*). We used *2D* Equations (8) and (10) for interaction MR$_I$ and *2D* Equation (13) for localisation MR$_L$. The modelled physical parameters from the MR$_I$ were the Coulomb interaction constant ($F_\sigma$) and the spin–orbit interaction energy ($\Delta E_{so}$) and from the MR$_L$ were dephasing length ($L_\phi$) and valley degeneracy factor ($n_v$). The dominant interaction effects in *NbN* were the Zeeman splitting energy ($g_e\mu_B H$) and the spin–orbit exchange interaction energy ($\Delta E_{so}$).

The Zeeman splitting energy is isotropic in an external magnetic field with electron g-factor ($g_e$ = 2.0). For in-plane external magnetic fields, scattering from the isotropic Zeeman splitting dominates the magnetotransport properties and scattering from the spin–orbit interaction energy is negligible. For out-of-plane external magnetic fields both scattering from isotropic Zeeman splitting and scattering from anisotropic dephasing and spin–orbit scattering influence the magnetotransport properties. We first modelled in-plane MR data by accounting only for the isotropic Zeeman splitting scattering and extracted the temperature-dependent Coulomb interaction constant ($F_\sigma$). $F_\sigma$ was the only physical parameter used for modelling in-plane MR with totally negligible spin–orbit interaction, i.e., $g_e\mu_B H(i/p) >> \Delta E_{so}(i/p)$. In addition, magnetoconductivity from interaction effects was very small. In-plane spin–orbit interaction energy $\Delta E_{so}(i/p)$ and in-plane dephasing length $L_\phi(i/p)$ could not be determined. $F_\sigma$ significantly increased with a decrease in temperature to $T_c$ (Figure 5). For magnetic oxide semiconductor thin films we observed that $F_\sigma$ decreased with a decrease in temperature [7]; however, for superconducting thin films we saw an increase in $F_\sigma$ with a decrease in temperature towards $T_c$. This could be interpreted in terms of a strong repulsive force experienced by the electrons around the Fermi energy when approaching $T_c$. This could possibly be due to the increase in density of electronic states around the Fermi level when approaching $T_c$ [20]. With an increase in the density of states around the Fermi level one would expect a stronger Coulomb repulsion among electrons. Studies by Zheng et al. [21] on the Coulomb repulsion between electrons forming Cooper pairs concluded that Coulomb interaction does not affect the temperature-dependent Cooper pair binding; therefore, the shift of $T_c$ is only marginally influenced by an enhanced Coulomb repulsion with an increasing density of states around the Fermi energy.

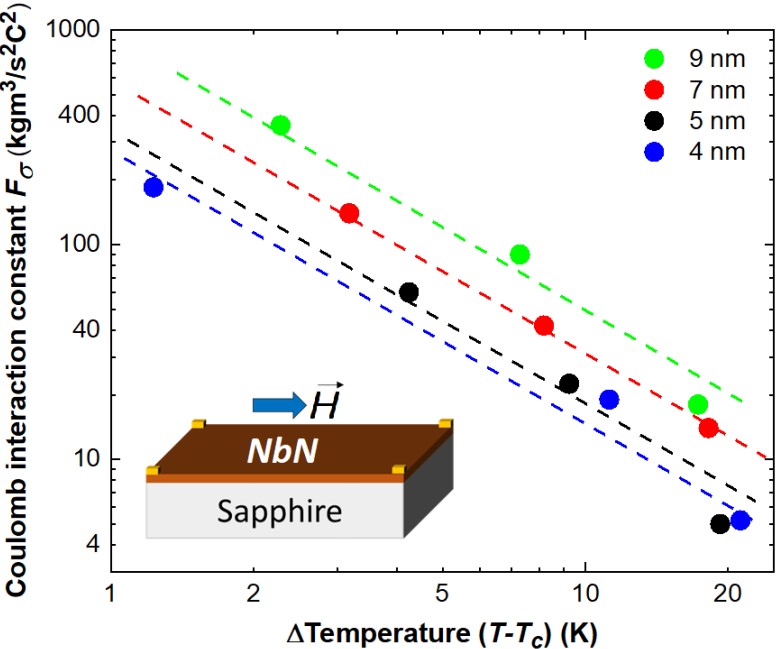

**Figure 5.** Temperature-dependent Coulomb interaction constant ($F_\sigma$) extracted from modelling in-plane MR data (open symbols in Figure 4). Scattered lines in the figure are only guide for the eye.

We modelled out-of-plane MR data by accounting for both the isotropic and anisotropic scattering and kept the Coulomb interaction constant ($F_\sigma$) as it had been extracted from modelling in-plane MR data (Figure 4). The temperature-dependent dephasing length ($L_\phi$), valley degeneracy factor ($n_v$), and spin–orbit scattering energy $\Delta E_{so}$ are listed in Table 1. First, 20 K and 30 K were modelled to extract the $\Delta E_{so}$ and $L_\phi$ from out-of-plane. Later, for temperatures approaching $T_c$, a good fitting model was only possible by increasing $n_v$. The effective electronic g-factor, $g_{eff}$ = 12 from *4d–4d*, *2s–4d*, and *2p–4d* interactions from $\Gamma_{12}$ near Fermi level [22], was used to model out-of-plane for spin–orbit interaction $\mu_{eff}$ in Equation (12), where $\mu_{eff} = g_{eff}\mu_B$. Similarly to significantly increased $F_\sigma$, a significant increase in $n_v$ (Table 1) just above $T_c$ hints towards the lifting of the degeneracy of electronic states around Fermi level in an external magnetic field, supporting the *BCS* theory [20]. The modelled range and temperature dependence of $\Delta E_{so}$ agreed with results reported by Shun-Tsung Lo et al. [9]. The modelled dephasing length ($L_\phi$) was in the range of the Ginzburg Landau coherence length ($\xi_{GL}$) reported by Chockalingam et al. [23] for electrons in a weak dephasing regime [24]. □

## 5. Conclusions

We analysed magnetoconductivity in ultrathin *NbN* films (*2D*) above a critical temperature ($T_c$) for four different thicknesses, namely 4 nm, 5 nm, 7 nm, and 9 nm, on sapphire substrates. The in-plane and out-of-plane magnetoresistance (reciprocal magnetoconductance) were modelled and discussed. The analysis approach from magnetic conducting oxide ultrathin films (*2D*) was used for *NbN* ultrathin films (*2D*) above $T_c$. Only the anisotropic *s–d* exchange interaction energy for oxide ultrathin films was replaced by the more general spin–orbit exchange interaction energy for *NbN* ultrathin films (2D) above $T_c$. We observed a significant increase in extracted Coulomb interaction constant ($F_\sigma$) and valley degeneracy factor ($n_v$) when the measurement temperature approached $T_c$. We reduced the fitting variables by incorporating the anisotropic spin–orbit interaction energy into the magnetocondutivity equations along with the isotropic Zeeman splitting energy in a weakly localised regime, which was traditionally analysed with spin–orbit scattering lengths or spin–orbit scattering time. We state that the combined analysis on in-plane and out-of-plane magnetoconductivity is a suitable approach to first extract $F_\sigma$ from in-plane measurement data and to extract the other parameters from out-of-plane MR with $F_\sigma$ from in-plane MR analysis. The MR analysis results could form the basis for further studies towards a deeper understanding of the transport mechanism in type-II superconducting *NbN* thin films at temperatures above $T_c$.

**Author Contributions:** S.V.V. worked on preparing the manuscript along and modelled and analysed the magnetotransport properties; S.V.L. worked on figures and analyzed Hall measurements; D.B. deposited the gold contacts for transport measurements and developed the methodology fother mangetotranpost measurements; D.B. and Z.L. conducted magnetotransport measurements; S.L. conducted the ultrathin *NbN* thin film sample preparation and assisted in developing the scientific approach in the manuscript; H.S. conceptualised the research, assisted in modelling analysis, contributed to the scientific approach, and gathered funds for the project. All authors have read and agreed to the published version of the manuscript.

**Funding:** The research was supported by a financial grant to D.B. and H.S. and they acknowledge support from the Deutsche Forschungsgemeinschaft (Grant Code: BU 2956/1-1 and SCHM 1663/4-1). S.V.V. and H.S. acknowledge funding from the Federal Ministry of Education and Research, APPA Photion (Code: 05P2021), and from the Office of Technology Assessment at the German Bundestag, TAB-AIF QHub (Code: 2021 FGI 0049).

**Institutional Review Board Statement:** Not applicable.

**Informed Consent Statement:** Not applicable.

**Data Availability Statement:** All the data needed to evaluate the conclusions are presented in this paper.

**Acknowledgments:** The authors thank Detlef Born for fruitful discussions as well as Mario Ziegler for support with *NbN* film deposition; both are colleagues from Leibniz IPHT Jena/Germany. We heartily acknowledge the support of the Open Access Publication Fund of the Thüringer Universitäts- and Landesbibliothek Jena/Germany.

**Conflicts of Interest:** The authors declare no conflict of interest.

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
