# Peer review of "Analysis of Low-Temperature Magnetotransport Properties of NbN Thin Films Grown by Atomic Layer Deposition"

_magnetochemistry, doi:10.3390/magnetochemistry8030033_

Round 1
Reviewer 1 Report
The in-plane and out-of-plane magnetoresistance of ultrathin NbN films grown by atomic layer deposition has been thoroughly studied. The data have been carefully modelled, and various physical parameters could be extracted by the combined analysis of in-plane and out-of-plane magneto-conductivity, which allowed, for example, for the distinction of interaction and localization effects.
The models introduced by the authors are very interesting and carefully explained and might be used in future analyses also by other research groups. I have not seen any technical or physical deficiencies.
Therefore, besides a few minor English flaws (mainly missing articles), I can recommend the paper for publication in the journal Magnetochemistry.
Author Response
we are thankful for considering our manuscript entitled “Low temperature magneto- and Hall resistance properties of thin NbN films grown by atomic layer deposition” by Sahitya V. Vegesna, Sai V. Lanka, Danilo Bürger , Zichao Li, Sven Linzen, and Heidemarie Schmidt for publication in the special issue "Magnetic and Transport Properties of Thin-Film Materials" after minor revision.
We appreciate the reviewers’ comments and are delighted to read their interest in the results of our work presented in the submitted manuscript. We have improved the English, mainly adding missing articles as suggested.
Reviewer 2 Report
In this manuscript, the authors analyzed magnetoconductivity in ultrathin NbN films with various thicknesses. By performing Hall and magnetoresistance measurements, together with appropriate fitting, the authors obtained various parameters such as Coulomb interaction constants, spin-orbit scattering lengths, localization lengths, and valley degeneracy factors. Moreover, the authors found that the combined analysis on in-plane and out-of-plane magnetoconductivity is able to obtain different parameters relevant to electronic transport. Overall, the story seems to be complete to me. I would suggest publication once the authors can address following points:
- Figure 1: It seems that for 4-nm NbN thin film, the superconducting transition is not complete. Is there any data below 8 K that can show a complete transition for it? If not, the authors should describe the reason for this incomplete superconducting transition, i.e., not low enough temperature/SIT transition etc.
- Lines 168-169: The authors claim “This could be possibly due to increase in density of electronic states around the Fermi level when approaching Tc”. This makes sense to me. But please clarify it more with one or two sentences with, i.e., BCS theory.
- The methods about doing Hall and MR measurements may go to Materials and Methods section instead of the Results section.
- Figure 3: For 7-nm film, it seems that both parameters are not monotonically changing with increasing temperature while there’s a kink when the temperature is increased from 20 to 30 K. Can the authors explain what happened here? If the standard deviation is available, please also indicate here in this figure.
Author Response
We are thankful for considering our manuscript entitled “Low temperature magneto- and Hall resistance properties of thin NbN films grown by atomic layer deposition” by Sahitya V. Vegesna, Sai V. Lanka, Danilo Bürger , Zichao Li, Sven Linzen, and Heidemarie Schmidt for publication in the special issue "Magnetic and Transport Properties of Thin-Film Materials" after minor revision.
We appreciate the reviewers’ comments and are delighted to read their interest in the results of our work presented in the submitted manuscript. We have revised the manuscript accordingly and list our response to every remark step by step in the following:
Comment 1:
Figure 1: It seems that for 4-nm NbN thin film, the superconducting transition is not complete. Is there any data below 8 K that can show a complete transition for it? If not, the authors should describe the reason for this incomplete superconducting transition, i.e., not low enough temperature/SIT transition etc.
Response 1:
Unfortunately, there are no data below 8 K to show complete transition for the 4-nm NbN thin film. The reason for this incomplete superconducting transition is described in the following:
For the 4 nm thick sample the transition is not completed down to 8 K. 8 K is the minimum achieved temperature inside the used Hall/MR setup, see figure 1. However, previous investigations carried out at lower temperatures and inside a magnetically shielded setup show a complete transition still for such small film thicknesses \cite{Linzen et al. 2017}. The incomplete transition is most probably due to magnetic stray fields arising from the measurement setup (< 50 Gauss) \cite{ Mudassar et al. 2020}.
We have incorporated the changes in the manuscript from lines 76-85 as follows:
We measured the temperature dependent sheet resistivity (ρxx) and determined the critical temperature (Tc) as 8.77 K, 10.75 K, 11.82 K, and 12.72K for a NbN thickness of 4 nm, 5 nm, 7nm and 9 nm, respectively (see Fig. 1). For the 4 nm thick sample the transition is not completed down to 8 K. 8 K is the minimum achieved temperature inside the used Hall/MR setup, see figure 1. However, previous investigations carried out at lower temperatures and inside a magnetically shielded setup show a complete transition still for such small film thicknesses \cite{Linzen et al. 2017}. The incomplete transition is most probably due to magnetic stray fields arising from the measurement setup (< 50 Gauss) \cite{ Mudassar et al. 2020}. For generalization, in this work we defined the critical temperature as the temperature where the sheet resistivity ρ(Tc) drops to 1 μΩcm.
Comment 2:
Lines 168-169: The authors claim “This could be possibly due to increase in density of electronic states around the Fermi level when approaching Tc”. This makes sense to me. But please clarify it more with one or two sentences with, i.e., BCS theory.
Response 2:
With increase in the density of states around the Fermi level one would expect a stronger Coulomb repulsion among electrons. Studies by Zheng et al. on the Coulomb repulsion between electrons forming Cooper pairs concluded that Coulomb interaction does not affect the temperature dependent Cooper pair binding. Therefore, the shift of Tc is only marginally influenced by enhanced Coulomb repulsion with increasing density of states around the Fermi energy.
We have incorporated the changes in the manuscript from lines 171-178 as follows:
This could be possibly due to increase in density of electronic states around the Fermi level when approaching Tc. With increase in the density of states around the Fermi level one would expect a stronger Coulomb repulsion among electrons. Studies by Zheng et al. on the Coulomb repulsion between electrons forming Cooper pairs concluded that Coulomb interaction does not affect the temperature dependent Cooper pair binding. Therefore, the shift of Tc is only marginally influenced by enhanced Coulomb repulsion with increasing density of states around the Fermi energy.
Comment 3:
The methods about doing Hall and MR measurements may go to Materials and Methods section instead of the Results section.
Response 3:
Thank you for the suggestion and we have changed the section as follows:
Line 61: 2. Results
Line 76: 2.1. Critical temperature à Line 75: 1.1. Critical temperature
Line 89: 2.2. Hall Resistance à Line 93: 1.2. Hall Resistance
Line 101: 2.3. Magnetoresistance à Line 105: 2. Results
Comment 3:
Figure 3: For 7-nm film, it seems that both parameters are not monotonically changing with increasing temperature while there’s a kink when the temperature is increased from 20 to 30 K. Can the authors explain what happened here? If the standard deviation is available, please also indicate here in this figure.
Response 3:
The measurement error is indicated in Fig. 3. It lies in the range from 3.24E17 cm-3 to 1.97E19 cm-3 for carrier concentration and from 0.8E-4 cm^2/Vs to 86E-4 cm^2/Vs for mobility.
The error has been calculated as follows:
First we fitted the slope of Hall resistance in Origin and determined the error_HR and the error in carrier concentration (n_error = n(slope)-n(slpoe+error_HR) ). The obtained carrier concentration n are in the order of 6E22 cm-3 and the error in carrier concentration n_error is in the order between 1E17 cm-3 and 1E19 cm-3 which is ca. 5-3 orders of magnitude smaller than the carrier concentration n. Similarly, mobility error is calculated (µ_error = µ (slope)- µ (slpoe+error_HR) ).
We have increased size of error bars in Fig. 3 for better visualization of measurement error and incorporated the following changes in Fig. 3.
Figure 3: (Left axis) Temperature dependent carrier concentration in units of cm−3 and (right axis) mobility in units of cm2/Vs of a 9, 7, 5 , and 4 nm thick NbN thin film on Sapphire above the critical temperature. The carrier concentration and mobility of bulk NbN is also indicated. The measurement error is indicated by error bars and lies in the range from 3.24E17 cm−3 to 1.97E19 cm−3 for carrier concentration and from 0.8E-4 cm2/Vs to 86E-4 cm2/Vs for mobility.